# Low-Frequency Ground Vibrations Generated by Debris Flows Detected by a Lab-Fabricated Seismometer

**DOI:** 10.3390/s22239310

**Published:** 2022-11-29

**Authors:** Ching-Jer Huang, Hsin-Yu Chen, Chung-Ray Chu, Ching-Ren Lin, Li-Chen Yen, Hsiao-Yuen Yin, Chau-Chang Wang, Ban-Yuan Kuo

**Affiliations:** 1Department of Hydraulic and Ocean Engineering, National Cheng Kung University, Tainan 70101, Taiwan; 2National Science and Technology Center for Disaster Reduction, New Taipei City 23143, Taiwan; 3Institute of Earth Sciences, Academia Sinica, Taipei City 11529, Taiwan; 4Monitoring and Management Division, Soil and Water Conservation Bureau, Council of Agriculture, Nantou City 540206, Taiwan; 5Institute of Undersea Technology, National Sun Yat-sen University, Kaohsiung City 804, Taiwan

**Keywords:** seismometer, geophone, debris flow, low-frequency ground vibration, frequency–response function, Aiyuzi Stream

## Abstract

A lab-fabricated ocean bottom seismometer was modified and deployed terrestrially to detect low-frequency (<10 Hz) ground vibrations produced by debris flows. A frequency–response test of the new seismometer revealed that it can detect seismic signals at frequencies of 0.3–120 Hz. Its seismic ground motion detection ability was investigated by comparing its measurements of seismic signals produced by rockfalls with those of a geophone. Two new seismometers were deployed at the Aiyuzi Stream, Nantou County, Taiwan, in September 2012. Seismic signals produced by two local earthquakes, two teleseisms, and three debris flows detected by the seismometer in 2013 and 2014 were discussed. The seismic signal frequencies of the local earthquakes and teleseisms (both approximately 1800 km apart) were 0.3–30 and <1 Hz, respectively. Moreover, seismometer measurements revealed that seismic signals generated by debris flows can have minimum frequencies as low as 2 Hz. Time-matched CCD camera images revealed that debris flow surge fronts with larger rocks have lower minimum frequencies. Finally, because the seismometer can detect low-frequency seismic waves with low spatial decay rates, it was able to detect one debris flow approximately 3 min and 40 s before it arrived.

## 1. Introduction

Overcultivation in mountainous regions leads to frequent debris flows. Debris flows and other slope disasters, such as landslides and rockfalls, are also exacerbated by the extreme weather associated with global warming. Debris flows are rapid, gravity-induced flows of a mixture of rocks, mud, and water [1,2]. Debris flows are usually accompanied by loud airborne sounds and violent ground vibrations [3]. Early warning systems to protect local residents from debris flows have been developed. These warning systems can be categorized as advanced warning and event warning systems [4]. Advanced debris flow warning systems are usually established on the basis of rainfall intensity and duration [5,6]; event warning systems are based on the detection of the seismic and acoustic signals produced by debris flows. Although event warning systems are more accurate than advanced warning system, they provide insufficient times for evacuating nearby residents. To develop a robust and effective event warning system for debris flows, easily detectable signals associated with debris flows must be identified. In the past two decades, various ground vibrations sensors, including hydrophones [7,8], geophones [9,10,11,12,13,14], seismometers [15,16,17,18,19,20], and fiber-optic sensors [21], have been used to detect and study the seismic ground motions produced by debris flows. The frequencies of such ground vibrations detected using conventional geophones are 10–100 Hz; those at the surge front and flow tail are mainly 10–30 and 60–80 Hz, respectively [11,22]. 

The characteristics of debris flow signals can be obtained by analyzing their seismic ground vibration signals. For example, mean debris flow velocities can be determined by calculating the time lags of the seismic signals detected at various seismic sensors [9,11,16,17,23]. Moreover, the amplitude of ground vibrations is related to the flow velocity, discharge, and grain size of debris flows [19,20,23,24,25].

Conventional geophones have a working frequency of approximately 10–250 Hz. By contrast, seismometers, such as the Streckeisen STS-2 (Streckeisen, Pfunger, Switzerland), have a wider range of 0.0083 Hz (120 s) to 100 Hz (0.01 s). The Streckeisen STS-2 has been adopted in global seismic networks, such as GEOSCOPE [26]. Modern seismometers installed with electronic sensors, amplifiers, and recording devices have an even wider operating frequency range [27,28]. Because the spatial decay rate of high-frequency seismic waves is substantially larger than that of the low-frequency waves, geophones can only detect local events, whereas seismometers can also detect remote events. Recently, Lai et al. [18] deployed seismometers to monitor debris flows and discovered that debris flows can produce low-frequency seismic signals at 5–10 Hz. Lai et al. [18] demonstrated that a debris flow warning system using a seismometer can provide 5 min of early warning for debris flows. Tiwari et al. [29] discussed the seismic characteristics of a debris flow caused by a rock–ice avalanche occurring in the Himalayan glaciers. Cook et al. [30] reported that the detection and early warning of catastrophic flow events can be achieved using a regional seismic monitoring system. Kogelnig et al. [12] reported that their geophone sensor detected a debris flow 50 s before it reached the sensor. The detection time depends on the signal–noise ratio, the sensitivity of the sensor, and the site and soil where the sensor is deployed.

Many studies have reported that debris flows generate not only seismic ground motions but also infrasonic sounds at 4–15 Hz [12,13,14,19,20,31,32]. Huang et al. [33] demonstrated that a rock falling on a channel bed produced both ground vibrations and airborne sounds, and the spatial decay rate of airborne sounds was significantly smaller than that of ground vibrations. Accordingly, debris flow events can also be detected by monitoring low-frequency airborne sounds. Furthermore, because of the relatively lower decay rate of airborne sounds compared with that of the seismic signals, infrasound sensor installation locations can be far from event sites.

Recently, both infrasonic and seismic sensors have been deployed concurrently to increase the reliability of debris flows detection [12,13,14,19,20]. Kogelnig et al. [12] deployed both geophone and infrasound sensors to monitor debris flows at the Lattenbach torrent (Tyrol, Austria) and the Illgraben torrent (Valais Canton, Switzerland). They reported that the dominant frequency of debris flow seismic signals at the Lattenbach torrent was centered at 17 Hz, whereas that of infrasonic signals was at 6 Hz. This result was consistent with those reported by Zhang et al. [31] and Chou et al. [32]. Schimmel and Hübl [13] and Schimmel et al. [14] demonstrated that accurate detection of debris flows and debris floods can be achieved with only one seismic sensor (geophone) co-located with one infrasound sensor if the signals are fed to a microcontroller running a detection algorithm. They proposed a method of estimating the peak discharge and total volume of debris flows based on infrasound data.

Marchetti et al. [19] and Belli et al. [20] presented a seismo-acoustic analysis of debris flow events in Illgraben, Switzerland. Infrasound data were collected by an infrasound array comprising four or five sensors; the seismic signals were recorded by a co-located seismometer (LE-3Dlite MkIII, Lennartz Electronic, Tübingen, Germany) with a measuring range of 1–100 Hz. Both the front velocity and peak discharge were positively correlated with both the infrasonic and seismic maximum root-mean-square amplitudes. The seismic and infrasound signals had peak frequencies of 8 and 5 Hz, respectively. They explained this spectral difference by suggesting that infrasonic and seismic waves are generated by different source processes that act at the flow free surface and at the channel bed, respectively.

According to the aforementioned studies, two sensor types are typically used to detect seismic ground motions generated by debris flows: high-frequency geophones and broadband seismometers. Both have advantages and disadvantages. Geophones are compact and inexpensive; however, they detect high-frequency seismic signals, which decay rapidly and therefore are only suitable for detecting local events and high-frequency components of teleseisms. By contrast, seismometers are larger, heavier, and more expensive but can monitor both local and remote events. Recent improvements of geophones and seismometers have overcome some of their respective drawbacks. For example, a new commercially available geophone, Raspberry Shake 3D (Raspberry Shake, Alto Boquete, Panama), can measure low frequencies; it achieves a range of 0.5–40 Hz at a weight of 0.6 kg. A new seismometer, LE-3Dlite MkIII (Lennartz Electronic, Tübingen, Germany), achieves a measuring frequency range of 1–100 Hz at a compact size and only 1.9 kg. Chen et al. [34] applied the Raspberry Shake 3D (RS3D) to detect the ground vibrations produced by an artificial debris flow. The RS3D could detect ground vibration signals with frequencies higher than 5 Hz. Incidentally, a local earthquake was detected during their field tests; its seismic frequency range was measured to be 0.5–40 Hz, constrained by the capability of the sensor. Similarly, Marchetti et al. [19] and Belli et al. [20] used LE-3Dlite to detect seismic signals generated by debris flows in Illgraben, Switzerland.

In this study, a lab-fabricated ocean bottom seismometer (OBS) developed by Wang et al. [35], called Yardbird OBS, was modified for on-land deployment to detect low-frequency seismic signals generated by debris flows. The goal of producing the modified Yardbird OBS, hereafter referred to as the Yardbird seismometer, was to obtain a sensor with both small size and acceptable response period bandwidth. The primary modification was to reduce the low end of the geophone’s frequency range from 4.5 to 0.3 Hz by incorporating a current-mirror circuit [35,36,37]. Because the seismometer can detect seismic waves with frequencies less than 10 Hz, it can measure the low-frequency band (<10 Hz) of ground vibrations generated by debris flows. Furthermore, because it can measure broadband signals, it might also detect earthquakes. In previous studies, commercially available seismometers have been employed to detect low-frequency ground vibrations [18,19,20]. In contrast to commercially available seismometers, lab-fabricated seismometers can be customized to measure various frequency ranges depending on the needs of the user. The results of this study could provide useful information and techniques for developing remote-sensing and early warning devices for various geophysical events.

First, a frequency–response test of the seismic sensor installed in the Yardbird seismometer was performed at the Taiwan Earthquake Research Center (TEC) to reveal the Yardbird seismometer’s measurement characteristics. The Yardbird seismometer’s ability to detect seismic ground motions was then tested by comparing its measurements of ground vibrations produced by rockfalls with those obtained using a geophone. After these tests, the seismometer was deployed to monitor debris flows at Aiyuzi Stream, Nantou County, Taiwan; the site is well-known to have frequent debris flow events. Finally, we present seismic data for various events, including debris flows, local earthquakes, and remote earthquakes, to verify the capability of the Yardbird seismometer for measuring low-frequency seismic signals and to determine whether it can be used to increase the emergency response time.

## 2. Lab-Fabricated Seismometer

The Yardbird OBS developed by Wang et al. [35] was modified for terrestrial deployment to detect low-frequency seismic signals generated by debris flows.

### 2.1. Yardbird OBS

In the Yardbird OBS, a 4.5 Hz SM-6 Geophone (Seis Tech, Xi’an, China) was used to pick up the seismic signals. An amplifier with negative input impedance feedback was used to overdamp the geophone such that the lower bound of the respond period is extended to several seconds. The amplifier is nonlinear; that is, the frequency response is not flat throughout the band of interest. To obtain the response curve, modified geophone sensors were tested using a vibration exciter in the desired frequency range.

Figure 1 presents an overview and the key elements of the Yardbird OBS. It comprises several components: sensors, a leveling device, a data logger, an anchor release mechanism, and surface beacons. The data logger, amplifier, and battery package were assembled in a 17” glass sphere (instrumentation housing in Figure 1), and the modified geophone and leveling device were placed in an alumina housing (sensor housing in Figure 1). Another 17” glass sphere was used to provide the buoyancy required to lift the system for retrieval after the release of the anchor (buoyancy housing in Figure 1). A detailed description of the Yardbird OBS’s construction is available in Wang et al. [35].

### 2.2. Yardbird Seismometer

In the Yardbird seismometer, a modified geophone sensor, leveling device, amplifier, and data logger were assembled in a water-tight cylindrical housing made of stainless steel; the battery package was placed in a separate water-tight container with an IP68 rating. The data logger’s analog-to-digital converter converted analog three-axis ground vibration signals into digital signals at a sampling rate of 240 Hz with a resolution of 24 bits. The digital signals were transmitted via RS-485 to the instrument house for data recording. 

Figure 2 presents the debris flow monitoring system in which the Yardbird seismometer is used to record seismic signals generated by debris flows. The Yardbird seismometer was installed on a concrete board that was confirmed to be horizontal by using a bubble level meter. The Yardbird seismometer measured the three-axis ground vibration velocities continually with a sampling rate of 240 Hz. The measured data were saved in a file (1860 kB) every 10 min. The relayed data were recorded as binary data by a computer at Shen-Mu Station. The data were then transferred to a server over the Internet.

### 2.3. Frequency Response of the Yardbird Seismometer

The frequency response of the Yardbird seismometer for measuring seismic signals was tested using a mechanical vibration exciter. The test was conducted at the TEC. Figure 3 presents the amplitude and phase of the frequency–response function (FRF) of the modified geophone. The FRF is a complex function that contains both an amplitude and phase; it is typically used to characterize the dynamic behavior of a structure. As indicated in Figure 3, at 0.3–120 Hz, the measured amplitude using the Yardbird seismometer agreed with the actual amplitude with a deviation of less than 10% (approximately ~1 dB); thus, the modified geophone is suitable for detecting ground vibration signals with frequencies of 0.3–120 Hz. Many geological events produce ground vibrations or seismic signals within this frequency range, such as earthquakes, landslides, debris flows, rockfalls, and torrential rivers.

### 2.4. Performance Comparison of an Yardbird Seismometer and A Geophone

The performance of the Yardbird seismometer for detecting ground vibrations was compared with that of a conventional geophone (GS-20DX, Geospace Technologies, Houston, TX, USA) in a field test. A rock weighing 13.8 kg was released from a height of 2 m and hit a dry channel bed (Figure 4). The Yardbird seismometer and geophone were close to each other, and the distance between the rock fall location and the instruments varied between 3.5 and 42 m. The rockfall was repeated once every 15 s and ten times. Because the Yardbird seismometer was sensitive to low-frequency signals, ambient noises with frequencies less than 2 Hz were filtered out using a third-order Butterworth filter to obtain clean rockfall-induced seismic signals. According to Hilbert et al. [38] and Saló et al. [39], the dominant frequencies of ground vibrations caused by rockfall are 10–150 Hz. Therefore, this filtering does not eliminate the rockfall signals.

Figure 5 presents the time-series data of the ground vibrations along the *x*-axis detected by the two sensors at a distance of 3.5 m from the impact. The upper and lower plots present the results obtained by the geophone and the Yardbird seismometer, respectively. In this study, the *z*-axis is perpendicular to the ground, the *x*-axis is parallel to the channel, and the *y*-axis is across the channel. The amplitudes of the rockfall-induced ground vibrations recorded with the two sensors are approximately the same (Figure 5), indicating that both sensors have similar performance.

The results for measuring distance increases of 14, 28, and 42 m are presented in Figure 6, Figure 7 and Figure 8, respectively. If the measuring distance was greater than 14 m, the geophone could not detect ground vibrations produced by the rockfall; however, the seismometer could detect seismic ground motion even at a distance of 42 m.

To examine the difference in the spectral behavior of the ground vibrations detected using the geophone and Yardbird seismometer, Figure 9a presents the time-series signals obtained from the fifth rockfall (Figure 5) detected using the geophone; Figure 9b presents their power spectral density (PSD) obtained using fast Fourier transform (FFT) in units of (m/s)^2^/Hz. The results corresponding to Figure 9a,b but detected using a Yardbird seismometer are presented in Figure 10a,b. A comparison of Figure 9 and Figure 10 reveals that although the time-series signals obtained from both sensors have approximately equal amplitudes, the spectral behavior is clearly different. The geophone signal (Figure 9) is at 20–80 Hz, and that of the Yardbird seismometer signal (Figure 10) is at 2–80 Hz. Thus, the Yardbird sensor measures low-frequency signals from 2 to 20 Hz and therefore is more sensitive than the geophone in the low-frequency band.

Based on the FRF (Figure 3) and the comparison of the ground vibration signals from the field test, the characteristics of the geophone and Yardbird seismometer were obtained. Table 1 presents a comparison of the geophone, the Yardbird seismometer, the broadband seismometer used in GEOSCOPE [26], and two commercially available seismic sensors, namely the Raspberry Shake 3D geophone and the LE-3Dlite seismometer.

## 3. Deployment of Yardbird Seismometer for Monitoring Debris Flows

After the field tests were completed, two Yardbird seismometers were deployed at the bank of the Aiyuzi Stream, Shen-Mu Village, Nantou County, Taiwan, to monitor debris flows. The Shen-Mu Monitoring Station was established in September 2012. We chose the Aiyuzi Stream as our research site because debris flows occur frequently there, and other debris flow monitoring instruments, such as wire sensors and CCD cameras, have been installed at the site by the Soil and Water Conservation Bureau (SWCB), Council of Agriculture, Taiwan. Measurements from these instruments can be used to verify the occurrence of debris flows. Figure 11a presents a map of the island of Taiwan, including Shen-Mu Monitoring Station; Figure 11b presents the locations of the two wire sensors, the CCD camera, and the two Yardbird seismometers. The two wire sensors were deployed at the same location at approximately 50 and 150 cm above the riverbed. The distance between the wire sensors and the CCD camera was 100 m, and that between the CCD camera and the upstream seismometer (ST1) was approximately 30 m. The two seismometers at ST1 and ST2 were placed 200 m apart, enabling us to estimate the debris flow velocity from the detected seismic signals.

Shen-Mu Village lies at the confluence of three streams: the Aiyuzi Stream, Huosa Stream, and the Chusui Stream. These three streams merge, forming the Hesha Stream. The bedrock in Shen-Mu Village mainly comprises sandstone with shale interlayers [40]. Due to frequent stream erosion and earthquakes, the scree on the surface layer is unstable, and debris flows are common.

## 4. Seismic Signals of Local and Distant Earthquakes Detected by the Yardbird Seismometer

The Shen-Mu debris flow monitoring station was established in September 2012 and has received seismic signals from several local and distant earthquakes. In this section, seismic signals obtained by the Yardbird seismometer of some local and distant earthquakes in 2013 and 2014 are discussed. The seismic signals of the two earthquakes occurring in 2013 were obtained before any debris flow signals were observed.

In order to reveal both temporal and spectral characteristics of seismic signals, this study determined the power spectral density (PSD) of the obtained signals by using short-time Fourier transform with a Hanning window. The PSD values are presented in spectrograms. The dimensionless PSD is given in decibels and is defined as follows:(1)$$\mathrm{PSD}\left[\mathrm{dB}\right]=10\times \log_{10}({P/P}_{o})$$
where P is the power of the signals obtained from spectrogram calculations performed in MATLAB with a unit of (m/s)^2^/Hz, and P_o_ is the reference value of signal power and is set to 1 (m/s)^2^/Hz. Accordingly, a positive PSD value indicates that the power of the signal is greater than that of the reference. Colors indicate the strength of PSD; red and blue represent stronger and weaker ground vibrations, respectively. The PSD values of the debris flow events investigated in this study were mostly −140 to −90 dB; therefore, the color code for the bars corresponded to this range. Because seismic signals caused by debris flows are typically 1–120 Hz, based on the Nyquist-Shannon sampling theorem [41], the sampling rate was set to 240 Hz.

### 4.1. Local and Distant Earthquakes Detection in 2013

An earthquake of Richter magnitude (M_L_) 6.24 occurred in Taiwan at 10:03:19 (local time (LT), UTC + 8) on 27 March 2013. Its epicenter was at Ren-Ai Township, Nantou County, Taiwan. Seismic signals were detected by the Yardbird seismometers deployed at Shen-Mu Station. Figure 12a presents the *z*-axis ground vibration time-series data detected by the upstream seismometer at ST1. The official measurement of the maximum velocity amplitude in the *z*-axis by the Central Weather Bureau of Taiwan (CWB) at Sun-Moon Lake Station and that by the upstream seismometer were 0.0629 and approximately 0.005 m/s, respectively. Because Sun-Moon Lake Station is much closer to the epicenter than Shen-Mu Station is, the ground vibration amplitude measured at Shen-Mu Station was smaller than that recorded at Sun-Moon Lake Station.

The distance between the epicenter and Shen-Mu Station is only 46 km; hence, the Yardbird seismometer detected the seismic signals of the local earthquake in a relatively wide frequency band of ≈0.3–30 Hz (spectrogram in Figure 12b). Figure 12c presents the spectrogram at 0–10 Hz, revealing the low-frequency characteristics of the seismic signals. The sudden increase in the PSD of the seismic signals (Figure 12b,c) is also a clear indication of the occurrence of the earthquake. Figure 12c indicates that the low-frequency signals (<2 Hz) remained for a long time; moreover, two additional seismic signals were observed at approximately 600 s. The CWB earthquake records indicate two aftershocks at M_L_ 3.9 and 3.7 after the main earthquake at Nantou. Accordingly, the two seismic signals appearing at approximately 600 s were attributed to these aftershocks. In contrast to those of the main earthquake, the seismic signals of these aftershocks were too small to be visible from the time-series data of Figure 12a; however, they were visible in the spectrogram.

A moment magnitude scale (M_W_) 7.0 earthquake occurred in China at 08:02:46 on 20 April 2013, causing numerous casualties. The earthquake’s epicenter was located in Lushan County, Ya-An City, Sichuan Province. The distance between Ya-An City and Nantou County is approximately 1870 km. The upstream seismometer at the Shen-Mu Station detected minor ground vibrations at 08:10, approximately 8 min after the Ya-An earthquake occurred. According to the CWB, no local earthquake event occurred in Taiwan near that time. Therefore, the seismic signals detected by the Yardbird seismometer were long-period seismic waves caused by the Ya-An earthquake. Figure 13a presents the time-series data of the seismic ground motions in the *z*-axis measured by upstream seismometer from 08:10:00 on 20 April 2013. Figure 13b,c present the spectrogram of these data with frequency ranges of 0–120 and 0–10 Hz, respectively. Figure 13a reveals that the maximum ground vibration velocity was only $3\times 10^{-4}\;m/s$; this small vibration would probably not be noticed by humans. Despite the attenuation of the high-frequency signals due to the long distance, the Yardbird seismometer still detected the low-frequency signals. The frequency range of the seismic signals is indiscernible in Figure 13b; however, the 0–10 Hz spectrogram reveals that the frequency band was less than 1 Hz (red). Suriñach et al. [42] reported that the frequency band of seismic signals caused by a teleseism is approximately 0.1–1.0 Hz. The results in Figure 13c are consistent with those of Suriñach et al. [42].

### 4.2. Local and Distant Earthquakes in 2014

Data measured for two earthquake events occurring in 2014 reveal that the Yardbird seismometer can detect low-frequency (<10 Hz) seismic waves.

A local earthquake of M_L_ 4.4 occurred in Taiwan at 09:39 (LT) on 14 July 2014. The epicenter was at Ren-Ai Township, Nantou County, near that of the local earthquake on 27 March 2013. Figure 14a presents the time-series data measured by the upstream seismometer from 09:35 to 09:44 on 14 July; Figure 14b presents the corresponding spectrogram. Figure 14a reveals that the sensor observed a foreshock that appeared at 09:36:27, 206 s before the main earthquake occurred at 09:39:53. Figure 14b reveals that the frequency range of the foreshock was 0.3–20 Hz and that of the main shock was 0.3–40 Hz. The PSD of the main shock for signals of 0.3–20 Hz was much larger than that of the foreshock. The continuous signal at 60 Hz is attributable to a nearby poorly isolated cable carrying alternating current.

The Ludan earthquake, with a magnitude of M_W_ 6.1 (estimated 6.5 by China), occurred at 16:30:10 (LT) on 3 August 2014, in Yunnan Province, China. The distance between Ludan County and Nantou County is approximately 1767 km. Figure 15 presents the seismic signals in the *z*-axis measured by the upstream seismometer from 16:30 to 16:45 on the same day. The sensor began to receive seismic signals at 16:34:12, approximately 4 min after the earthquake occurred. The velocity amplitude of the background noise was approximately $0.8\times 10^{-5}\;m/s$; the maximum velocity amplitude of $2.6\times 10^{-5}\;m/s$ was detected at 16:40. The frequencies of the detected seismic signals were 0.3–1.0 Hz. On the basis of the frequency–response curve of Figure 3, signals with a frequency of <0.3 Hz were treated as noise. 

The results presented in Figure 12, Figure 13, Figure 14 and Figure 15 reveal that the Yardbird seismometer can detect low-frequency (<10 Hz) seismic ground motions. The goal of this study was to determine whether the lab-fabricated seismometer could detect low-frequency ground vibrations produced by debris flows; hence, a detailed analysis of the seismic signals from earthquakes is beyond the scope of this study.

## 5. Debris Flow Events

Debris flows in Taiwan are often triggered by torrential rains during the plum rain season (monsoon, May–June) and typhoons (mainly July–September).

### 5.1. Debris Flow Triggered by Torrential Rains in 2013 and 2014

A precipitation event with a rainfall duration of 3 h and a cumulative rainfall of 25 mm occurred at Shen-Mu Village on 30 May 2013. The peak rainfall intensity was 5.5 mm in 10 min (Figure 16). The lower wire sensor set up by the SWCB at headwaters cut off at 15:19:05, and the upper wire sensor cut off at 15:19:27. Images from the CCD camera, which was installed near the upstream seismometer, reveal that the precipitation event triggered small debris flows along the Aiyuzi Stream (Figure 17). The lower wire sensor is thought to have been cut off by the surge front of the newly formed debris flow; as the rainfall continue to accumulate, the surge of a larger debris flow then cut off the upper wire sensor.

During the debris flow, the downstream seismometer was out of order; accordingly, Figure 18 only presents the time-series data and spectrogram of the ground vibrations in the *z*-axis measured by the upstream seismometer at the Aiyuzi Stream from 15:00 to 16:00 (LT) on 30 May 2013. The PSDs in Figure 18 were calculated using Equation (1). Figure 18 reveals that the leading signal at approximately 12 Hz appeared at 15:16:04 when the accumulated rainfall reached 17 mm. Thus, the ground vibrations measured at this time were attributed to the stream flow. At approximately 15:20:00, as the precipitation continued, the bandwidth of the seismic signals increased exponentially (white arrow in Figure 18), ranging from 7 to 60 Hz; the PSDs of the seismic signals also increased. These increases were identified as typical phenomena associated with the onset of a debris flow [11,42,43,44,45]. By contrast, Figure 12 and Figure 14 reveal that the PSDs of earthquake-induced seismic signals are characterized by an abrupt increase from ambient noise. This difference in the temporal evolution of the bandwidth and PSD of the seismic signals may distinguish debris flow and earthquake events. The occurrence of this debris flow was verified by CCD camera images and the broken wire sensors. Figure 17 presents an image taken by the CCD camera at 15:19:26 on 30 May 2013. Figure 18 reveals that this image was taken close to the moment that the surge front of the debris flow passed by the upstream seismometer. The stone indicated by the red arrow was estimated to have a 0.5 m diameter. The seismic signals of this debris flow with a lowest frequency of 11 Hz at 15:19:26, as shown in Figure 18, may have been related to the movement of these stones.

Moreover, Figure 18 reveals that the upstream seismometer received intense seismic signals from 15:23 to 15:42. The signals occurring between 15:23:30 and 15:31:21 have a frequency range of 4–100 Hz with a greater PSD from 10 to 30 Hz (red); the signals that occur between 15:31 and 15:42 have a roughly similar but slightly higher frequency range of 6–100 Hz.

The front surge of a typical debris flow contains numerous large boulders. Based on geophone measurements, Okuda et al. [22] and Huang et al. [11] suggested that the large boulders generate the ground vibrations at lower frequencies (10–30 Hz). However, the debris flow seismic signals measured by our sensor capable of detecting signals at lower frequencies (0.3–120 Hz) revealed that ground vibration frequency can be as low as 4 Hz. In several recent studies [18,19,20], researchers have detected debris flow seismic waves by using seismometers; they reported that debris flows can produce low-frequency (<10 Hz) seismic signals. This finding is consistent with their results.

After 15:31, the minimum frequency increases from 4 to 6 Hz, and the strength of the seismic signal PSDs at 10–30 Hz also decreased. This change may be due to a decrease in the number of large boulders in the debris flow. After the debris flow surge passed the sensor at 15:42, the debris flow wake, which contains high water contents, silt, and fine sand, produced weaker ground vibrations with frequencies of 10–60 Hz.

During the monsoon season in 2014, torrential rainfall that happened on 20 May induced a debris flow. Figure 19 presents a chart of precipitation from 09:00 to 23:00 on 20 May measured every 10 min by the rain gauge at Shen-Mu Station. The accumulated rainfall during this period was 155.5 mm. The peak rainfall was 13 mm in 10 min and occurred at 12:50 PM. Figure 20a presents the time-series data of the ground vibrations in the *z*-axis measured by the upstream seismometer at the Aiyuzi Stream from 12:50 to 13:05; Figure 20b presents the spectrogram of these signals. Figure 20a,b reveal that following the peak rainfall at approximately 12:53, the river flow induced ground vibrations with frequencies ranging of 10–30 Hz. At approximately 12:55:30, the upstream seismometer detected the arrival of the debris surge front, which produced typical seismic signals at 4–80 Hz. These measurements revealed that, at the onset of the debris flow, both the PSD strength and the frequency range of the seismic signals increase rapidly.

Figure 21 presents an image taken by the CCD camera deployed at the Aiyuzi Stream at 12:54:19 on 20 May 2014. This image was taken close to the arrival time of the surge front of the debris flow at 12:55:30. The stone indicated by a red arrow has a diameter of approximately 2 m; the motion of this stone may have generated the low-frequency seismic signals at 4 Hz.

### 5.2. Debris Flow during Typhoon Soulik in 2013

Typhoon Soulik made landfall in northeastern Taiwan on 13 July 2013 at daybreak and induced a strong southwest airflow, which brought heavy rainfall to the mountainous area of Taiwan. Figure 22 presents the 10 min precipitation chart produced from measurements at Shen-Mu Station from 18:30 on 12 July to 12:00 on 13 July. The cumulative rainfall during this period was 499.5 mm.

Both the upstream and downstream seismometers received seismic signals caused by a debris flow during the typhoon. Figure 23 presents the seismic signals in the *z*-axis received by the upstream seismometer from 18:00 on 12 July to approximately 07:00 on 13 July 2013. Figure 23a presents the time-series data of the ground motion velocity, and Figure 23b presents the corresponding spectrogram. Figure 23c,d present spectrograms of the seismic signals from 01:00 to 02:00 and from 06:51 to 06:58, respectively, on 13 July. Figure 23b reveals that the sensor began to receive seismic signals at 15–30 Hz at approximately 22:00 on 12 July. After 23:00, these signals became stronger, with larger PSD values and a wider frequency range of 10–50 Hz. After 00:00 on 13 July, the 15–30 Hz signals had lower PSD values than those measured in the previous hour (23:00–24:00).

Based on the results in Figure 18 and Figure 20 and the literature [46,47], the seismic signals recorded from 22:00 on 12 July to 01:00 on 13 July were attributed to stream flows with varying discharges. After approximately 01:00 on 13 July, both the PSD and frequency range of the signals increase. This change is labeled as “first ground vibration” in the figure. The upstream seismometer at ST1 was damaged by the debris flow at approximately 07:00 on 13 July; thus, no further signals were recorded. The damage to the upstream seismometer by debris flow at 06:57 was verified by a CCD camera image (Figure 24) and by the rupture of the nearby wire sensors.

Figure 23c,d present the spectrogram of the signals acquired from 01:00 to 02:00 and from 06:51 to 06:58 on 13 July. Figure 23c reveals that the frequency range widens gradually from 12–40 Hz at 01:00 to 4–80 Hz at 01:22; subsequently, it decreases to 6–60 Hz at 01:34 and then remains roughly constant until 02:00. Temporal frequency variations indicate that the stream flow with increasing discharges (due to strong precipitation from 01:00 to 02:00, Figure 22) developed into a debris flow containing large boulders, resulting in the lowest frequency decreasing to 4 Hz at 01:22. Identifying the exact time that debris flow occurred from Figure 23c is difficult because no sharp surge front can be discerned. However, based on the PSDs in Figure 23b, the first debris flow surge appeared at approximately 01:18.

Figure 23d reveals that the frequency band of the signals from 06:51 to 06:53:30 is approximately 8–80 Hz; that from 06:53:30 to 06:56:50 is approximately 4–100 Hz. After 06:56:50, the PSDs of the signals increase, and the frequency range widens to 2–120 Hz, indicating that a strong debris flow had formed. After 06:57:12, no more signals were received, indicating that this debris flow destroyed the upstream seismometer. Figure 23d reveals that this second debris flow surge had a sharp front and passed the sensor at 06:57:00. This debris flow was attributed to the heavy precipitation from 06:00 to 07:00 on 13 July as indicated in Figure 22. Figure 24a,b present images of the debris flow taken by the CCD camera at 06:57:03 and 06:57:28 on 13 July 2013, respectively. The image in Figure 24a reveals that the debris flow contained two large rocks of approximately 3–5 m in diameter. The movement of these rocks may be the source of the low-frequency ground vibrations with a minimum frequency of 2 Hz. The image in Figure 24b indicates that the debris flow surge passed the CCD camera. 

This analysis suggests that heavy rain at 01:00–02:00 induced the first debris flow surge occurring at 01:18 (“First ground vibration” in Figure 23b). The first debris flow surge elapsed at 03:00 because the precipitation from 02:00 to 03:00 was light. After 03:00, further heavy rain induced another debris flow surge. The debris flow surge that occurred at 06:57:00 was induced by heavy rainfall from 06:00 to 07:00. This debris flow surge eventually damaged the upstream seismometer at 06:57:12.

Seismic signals corresponding to the same event in the *z*-axis measured by the downstream seismometer at ST2 from 18:00 on 12 July to 15:00 on 13 July 2013 are presented in Figure 25a. In Figure 25a, no additional seismic signals were received after approximately 14:44:20 because the downstream seismometer was eventually damaged due to successive debris flow surges. Figure 25b–d present spectrograms of signals collected over various times. Figure 25a,b reveal that the sensor began to receive seismic signals at approximately 21:15 on 12 July; the rainfall rate at that time was approximately 1.7 mm per 10 min (Figure 22). The majority of the signals measured from 21:15 on 12 July to 01:00 on 13 July were 8–70 Hz and were attributed to the stream flow.

At approximately 01:00 and 07:00, the frequency range widened and the PSD increased rapidly. Spectrograms of the signals from 01:00 to 02:00 and from 06:55 to 07:02 are detailed in Figure 25c,d, respectively. Figure 25c reveals that the frequency ranges of signals widened from 8–70 Hz after 01:00 to 2–110 Hz at 01:22. This latter range is a typical seismic signal of debris flow. Accordingly, the stream flow developed to a debris flow at 01:20–01:22. As in Figure 23c, this debris flow surge also lacks a sharp front; thus, identifying the exact arrival time of the debris flow from these data is also difficult. The arrival time of the first debris flow was estimated to be 01:21, according to Figure 25b. Figure 25d indicates that the second debris flow surge occurred at 06:57:17. The scale of the second surge was larger than the first surge at 01:21; the PSDs indicate that the generated ground vibrations were stronger, and they had a wider frequency range and a sharper front. The frequency range extended to both the higher and lower bands. The lowest frequency was 2 Hz. 

A comparison of the spectrograms obtained by the two seismometers reveals that the first debris flow surge occurred at ST1 at approximately 01:18 (Figure 23b) and at ST2 at approximately 01:22 (Figure 25b). Because the debris flow surge lacked a sharp front; its flow velocity is difficult to determine from the detected seismic signals. By contrast, the second debris flow surge was large with a sharp front; it arrived at the upstream and downstream sensors at 06:57:00 and 06:57:17, respectively. The distance between the two sensors was 200 m, and the debris flow crossed this distance in 17 s; hence, the mean flow velocity was 11.8 m/s.

### 5.3. Discussions

In this study, we demonstrated that the lab-fabricated Yardbird seismometer with a working frequency range of 0.3–120 Hz can detect debris flows and both local and distant earthquakes. On the basis of the measured low-frequency seismic signals (<10 Hz), debris flows were determined to generate ground vibrations as low as 2 Hz; such signals cannot be measured using conventional geophones. Because the spatial decay rate of low-frequency seismic waves is lower than that of high-frequency waves, these low-frequency signals can be detected earlier. A more detailed examination of the seismic signals validates this fact. Figure 23d reveals that the second debris flow surge was detected by the upstream seismometer at 06:57 on 13 July 2013; however, the sensor received low-frequency signals (<4 Hz) at approximately 06:53:20, 3 min and 40 s before the debris flow surge arrived. This early detection of debris flow is critical for effective disaster mitigation. In addition, the velocity of the debris flow was estimated to be 11.8 m/s; thus, the debris flow was approximately 2596 m away from the upstream seismometer when it was first detected at 06:53:20. Figure 26 reveals that this upriver location is near an area with scree deposits and may have been the source site of the debris flow. The sources of this scree are rock and gravel from nearby bare scarps. Because the flow source was estimated on the basis of the averaged debris flow velocity between the two seismometers, which were both deployed far from the flow source, this location is only a rough estimate.

The surge front of a debris flow contains large rocks [46], and the movement of rocks rubbing and colliding against the riverbed induces seismic ground vibrations [8,11]. Based on these references and the results in Figure 17, Figure 18, Figure 20, Figure 21, Figure 23 and Figure 24, the low-frequency seismic signals of debris flow were attributed to the motion of large rock blocks in the flows. Table 2 summarizes the relation between the observed rock sizes and the lowest detected frequency at the time when the rock images were obtained. The lowest seismic signal frequency appears to be inversely correlated with the size of rocks in the surge front; flows with larger rocks have a smaller lowest frequency. The detection of these low-frequency signals in debris flows by using seismometers has also been reported by other researchers [14,18,19,20]. Saló et al. [39] investigated the seismic energy generated by single-block rockfalls with masses ranging between 466 and 13,581 kg. For most impacts, the dominant frequency band of the seismic signals was 5–120 Hz, and the most energetic frequencies were approximately 50 Hz. This frequency range is approximately the same as that observed for debris flows.

In addition to the seismic signals from debris flows and earthquakes, we detected ground vibration signals from the river flows with a frequency bandwidth of 8–70 Hz. This frequency bandwidth is consistent with the results obtained by Burtin et al. [47,48] of 2–25 and 2–80 Hz, respectively.

Many geophysical events produce seismic waves with frequencies ranging from 0.3 to 120 Hz, which lie within the working frequencies of the Yardbird seismometer. These events include earthquakes, volcanic eruptions, rockfalls, snow avalanches, debris flows, pyroclastic flows, and landslides, and could be identified using detected signals based on their characteristic frequency ranges.

Table 3 summarizes the dominant frequency ranges of seismic signals produced by various events. The frequency range depends on the scale of an event, the site of the sensor, and other variables. The results in Table 3 are collected from selected references and are certainly not exhaustive. Recently, Cook and Dietze [49] provided a detailed discussion of the characteristics of seismic signals produced by geomorphic processes, such as rockfalls, river flows, floods, landslides, and earthquakes.

Moreover, Suriñach et al. [42] stated that seismic signal spectrograms differ between geophysical events; those generated by mass movements on the surface, such as debris flows, avalanches, pyroclastic flows, and landslides, have a gradual increase in the frequency and amplitude of seismic signals, whereas earthquakes, rockfalls, and explosions have an abrupt increase. Accordingly, both differences in the frequency ranges and the temporal characteristics of seismic signals could be used to distinguish geophysical events.

## 6. Conclusions

A lab-fabricated OBS was modified and deployed terrestrially to monitor seismic signals produced by debris flows. This seismometer was referred to as the Yardbird seismometer. A frequency–response test of the seismometer performed at the TEC revealed that it can detect seismic ground motion with frequencies ranging from 0.3 to 120 Hz.

Two Yardbird seismometers were deployed in September 2012 along the Aiyuzi Stream, Shen-Mu Village, Nantou County, Taiwan, and captured seismic signals produced by two local earthquakes and two teleseisms. The dominant frequency ranges of the seismic signals caused by the local earthquakes and teleseisms (at approximately ~1800 km) were 0.3–30 and ≤1 Hz, respectively.

The Yardbird seismometer also detected three debris flow events, which were verified with rainfall rate data, wire sensors, and CCD camera images. These three debris flow events were small, medium, and large. An examination of the characteristics of the corresponding seismic signals revealed that low-frequency signals were detected by the Yardbird seismometer; the minimum frequency of debris flow seismic signals can be as low as 2 Hz. This frequency cannot be measured using a conventional geophone. These low-frequency components of debris flow seismic signals are consistent with those reported by other researchers.

The spatial decay rate of low-frequency seismic waves is lower than that of high-frequency waves; hence, the seismometer received low-frequency signals of the debris flow that occurred on 13 July 2013 approximately 3 min and 40 s before the debris flow surge arrived at the sensor. This early detection of debris flow enabled by the low-frequency sensing capability of the Yardbird seismometer is critical for disaster mitigation.

Because both the upstream and downstream seismometers received complete seismic signals of the debris flow that occurred on 13 July 2013 (before they were damaged), the mean velocity of the debris flow could be determined to be 11.8 m/s. Moreover, the source location of the debris flow was estimated based on the time that the low-frequency signals were first detected.

The detected seismic signals and the CCD camera images indicate that the low frequencies of debris flow seismic signals may be caused by the motion of large rocks in the debris flows; debris flows with larger rock sizes had lower minimum frequencies.

Criteria, namely the frequency range and temporal characteristics of the seismic signals, for distinguishing debris flow events from other geophysical events, such as earthquakes, rockfalls, landslides, and snow avalanches, were briefly discussed.

Overall, the results of this study reveal that the Yardbird seismometer is a cost-effective sensor for both debris flow research and implementing early warning systems for geophysical events. Future studies could further examine its performance for detecting other events, such as rockfalls, landslides, or earthquakes.

## Figures and Tables

**Figure 1 sensors-22-09310-f001:**
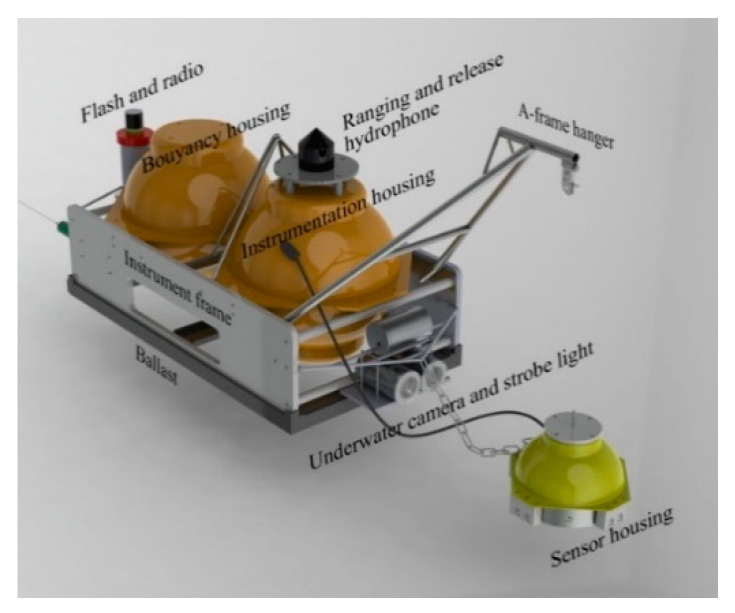
Overview and key elements of the Yardbird OBS.

**Figure 2 sensors-22-09310-f002:**
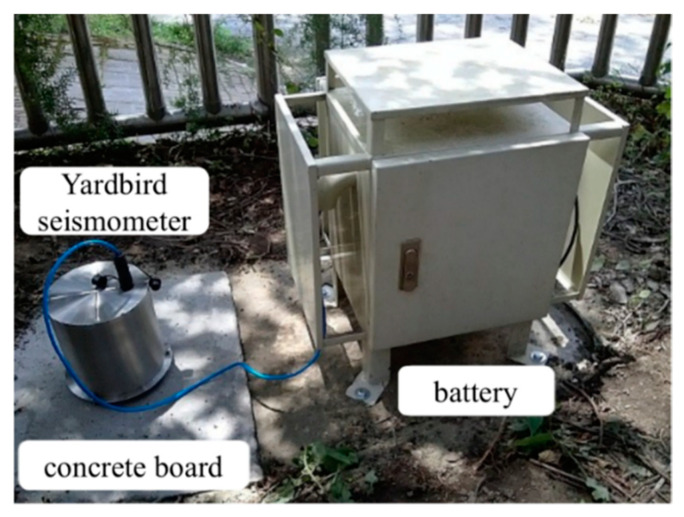
The debris flow monitoring system. The seismometer was assembled in a stainless housing for terrestrial deployment for detecting ground vibrations generated by debris flows. The battery package is placed in a separate water-tight container. The seismometer was installed on a concrete board, which has been tested to be horizontal.

**Figure 3 sensors-22-09310-f003:**
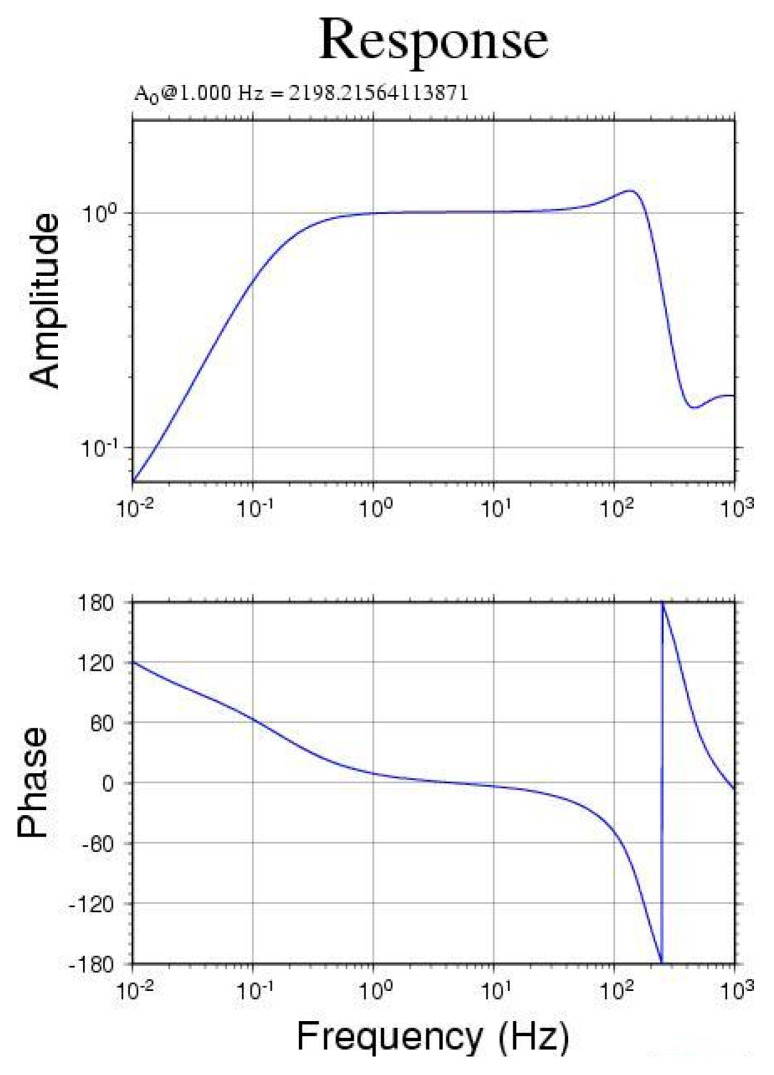
Amplitude and phase of the frequency–response function for the Yardbird seismometer. The test was conducted using a mechanical vibration exciter at the TEC.

**Figure 4 sensors-22-09310-f004:**
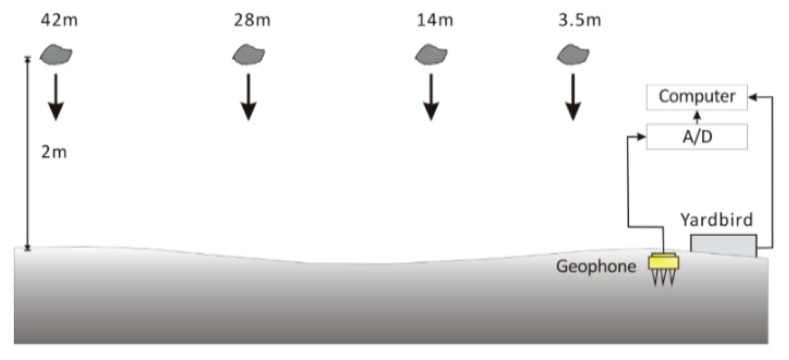
Detection of ground vibrations produced by rockfall at various distances from the Yardbird seismometer and geophone.

**Figure 5 sensors-22-09310-f005:**
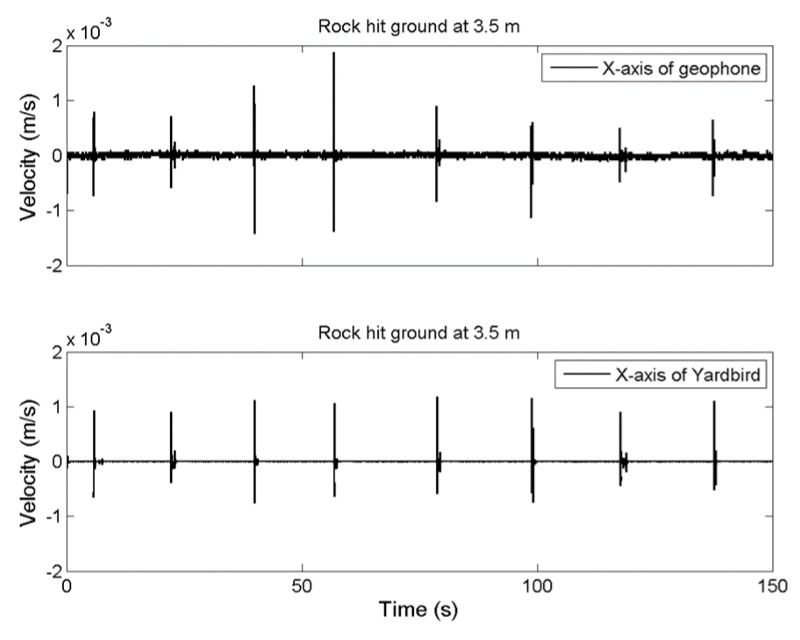
Time series of ground vibrations in the *x*-axis produced by a 13.8 kg rock released from a height of 2 m and detected by a geophone (**upper**) and a Yardbird seismometer (**lower**). The distance between the rock impact location and the measurement location is 3.5 m.

**Figure 6 sensors-22-09310-f006:**
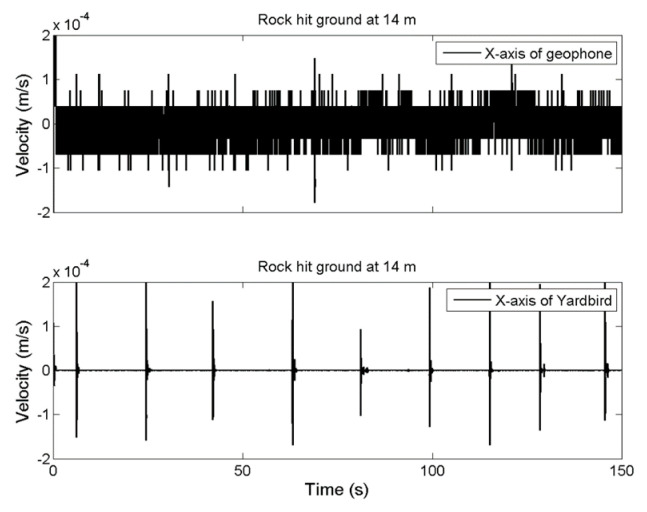
Ground vibration time-series data for the rockfall measured by the geophone (**upper**) and seismometer (**lower**) at a distance of 14 m.

**Figure 7 sensors-22-09310-f007:**
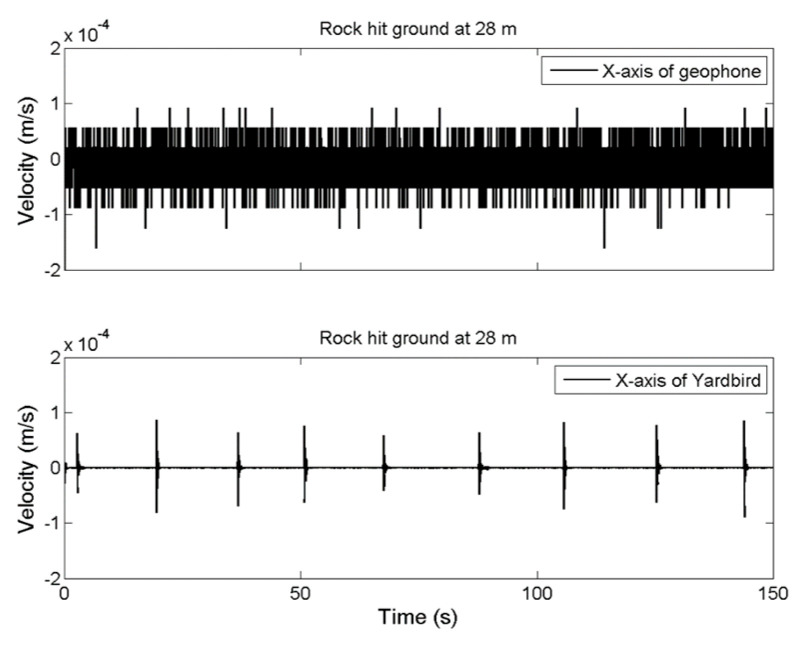
Ground vibration time-series data for the rockfall measured by the geophone (**upper**) and seismometer (**lower**) at a distance of 28 m.

**Figure 8 sensors-22-09310-f008:**
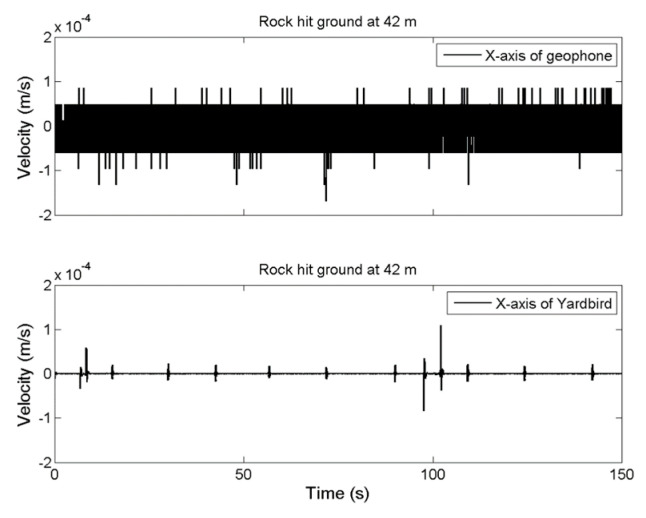
Ground vibration time-series data for the rockfall measured by the geophone (**upper**) and seismometer (**lower**) at a distance of 42 m.

**Figure 9 sensors-22-09310-f009:**
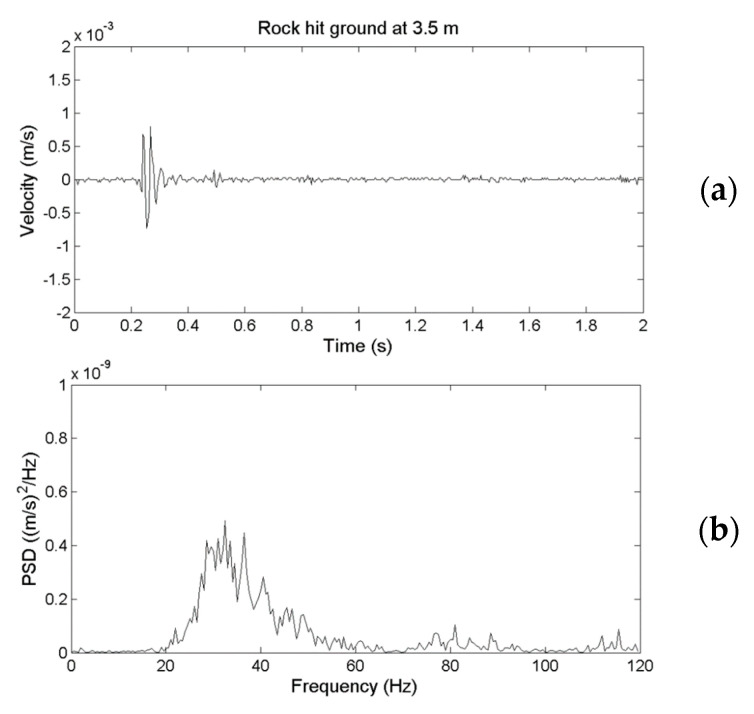
(**a**) Temporal and (**b**) spectral behavior of ground vibrations produced by the rockfall at a distance of 3.5 m from the measuring location detected by a geophone.

**Figure 10 sensors-22-09310-f010:**
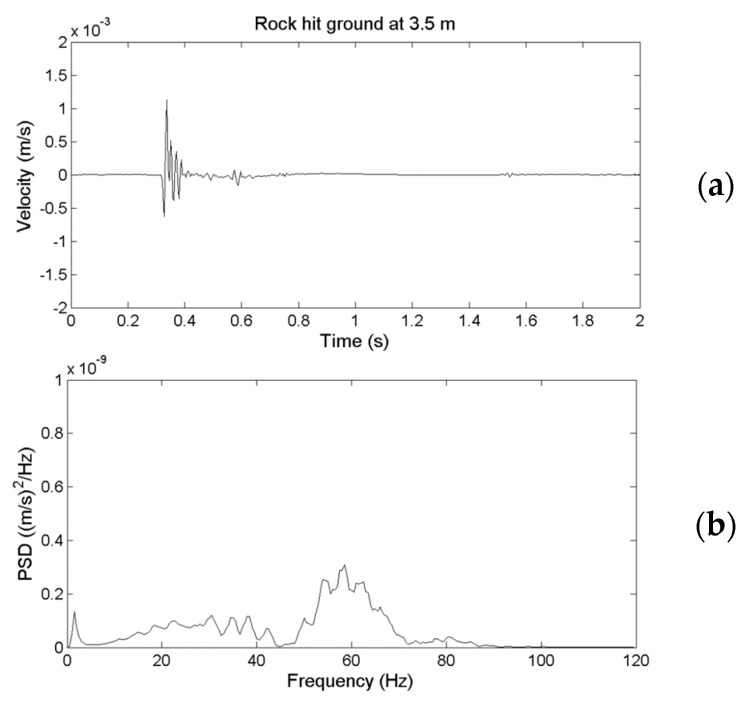
(**a**) Temporal and (**b**) spectral behavior of ground vibrations produced by the rockfall at a distance of 3.5 m from the measuring location detected by the Yardbird seismometer.

**Figure 11 sensors-22-09310-f011:**
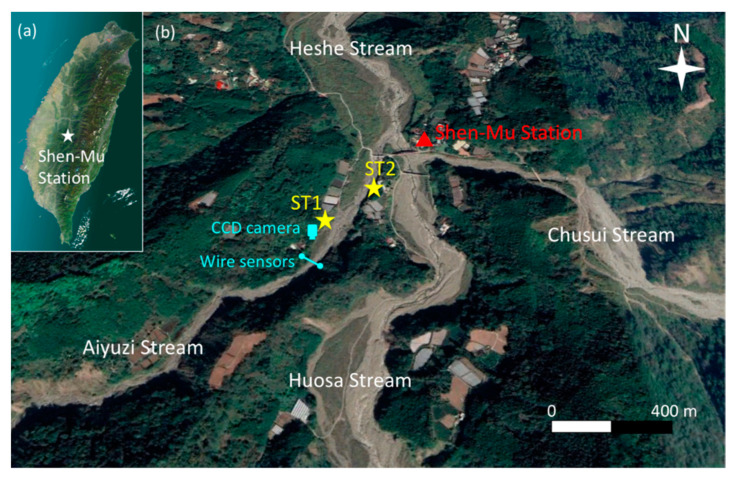
(**a**) Map of Taiwan and location of Shen-Mu Station. (**b**) Locations of the wire sensors, CCD camera, and Yardbird seismometers deployed at ST1 and ST2 along the Aiyuzi Stream in the Shen-Mu Station. (From Google Earth).

**Figure 12 sensors-22-09310-f012:**
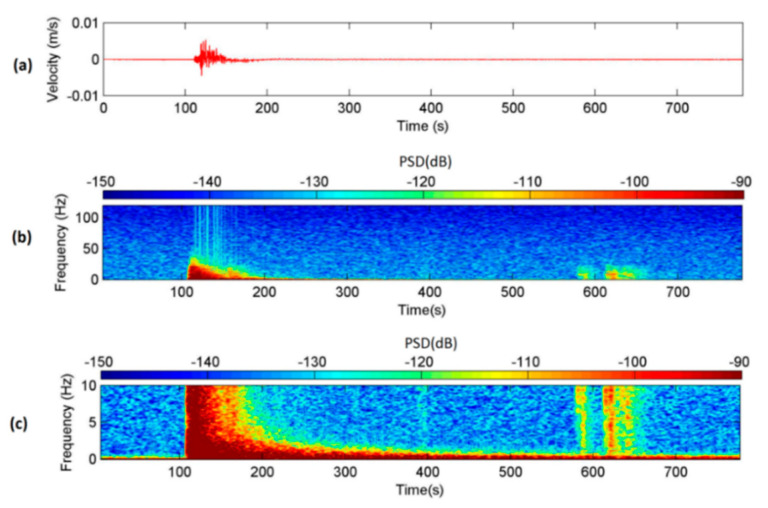
Seismic signals in the *z*-axis detected by the seismometer at ST1 and produced by the Nantou earthquake occurring at 10:03:19 (LT) on 27 March 2013. (**a**) Time-series of the ground vibration velocity. (**b**) Spectrogram of the seismic signals from 0 to 120 Hz. (**c**) Spectrogram of the seismic signals from 0 to 10 Hz.

**Figure 13 sensors-22-09310-f013:**
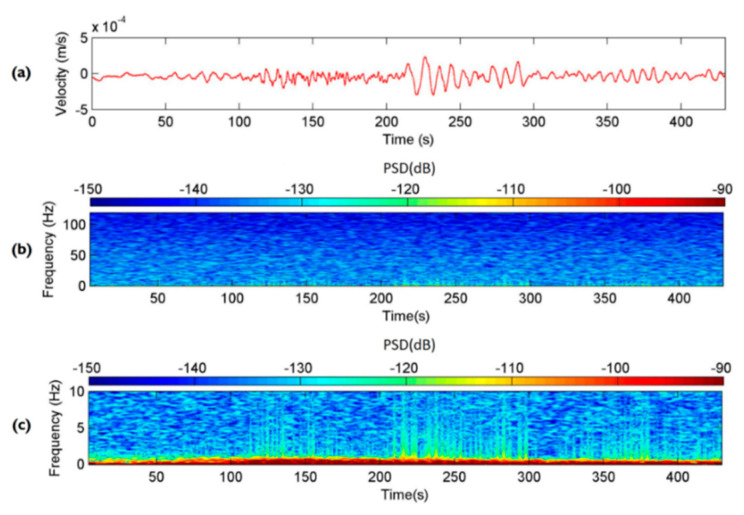
Seismic signals attributed to the Ya-An earthquake in the z-axis detected by the upstream seismometer from 08:10:00 on 20 April 2013. (**a**) Time series of the velocity of ground vibrations. (**b**) Spectrogram of the seismic signals from 0 to 120 Hz. (**c**) Spectrogram of the seismic signals from 0 to 10 Hz.

**Figure 14 sensors-22-09310-f014:**
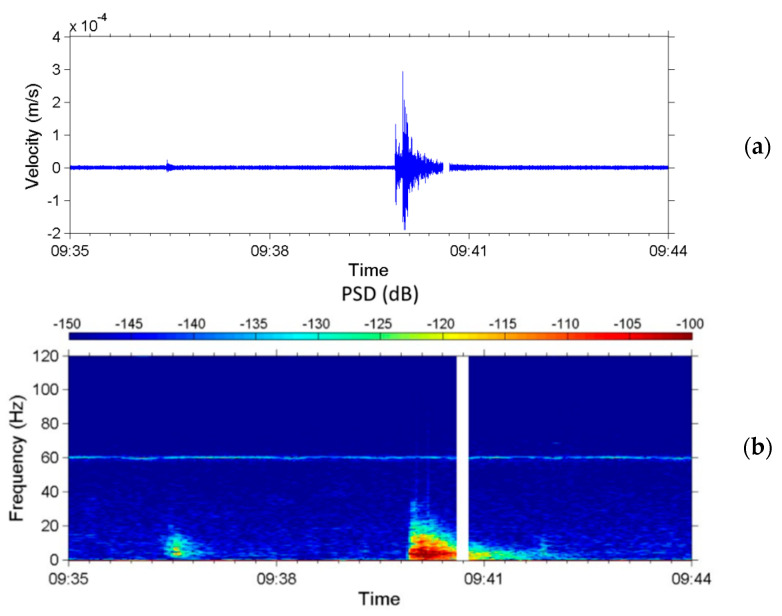
Seismic signals attributed to the Nantou earthquake in the z-axis occurring on 14 July 2014. (**a**) Time-series of the velocity of ground vibrations and (**b**) spectrogram of the seismic signals from 0 to 120 Hz.

**Figure 15 sensors-22-09310-f015:**
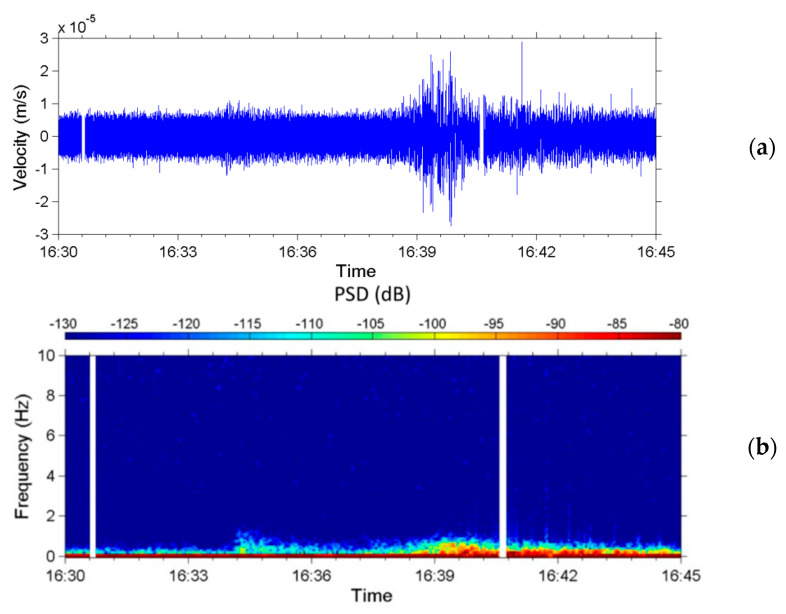
Seismic signals attributed to the Ludan earthquake in the z-axis occurring on 3 August 2014 in Yunnan Province, China. (**a**) Time-series of the velocity of ground vibrations and (**b**) spectrogram of the seismic signals from 0 to 10 Hz.

**Figure 16 sensors-22-09310-f016:**
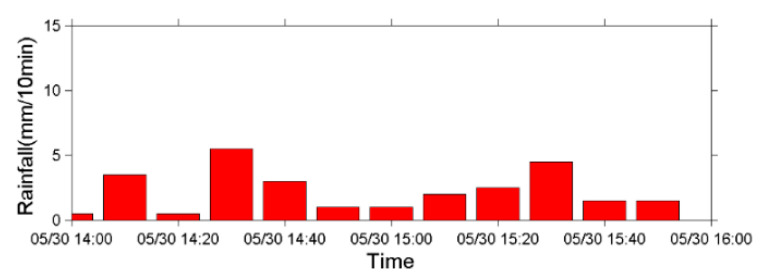
Precipitation chart in 10 min intervals produced based on rain gauge measurements from the Shen-Mu Station on 30 May 2013.

**Figure 17 sensors-22-09310-f017:**
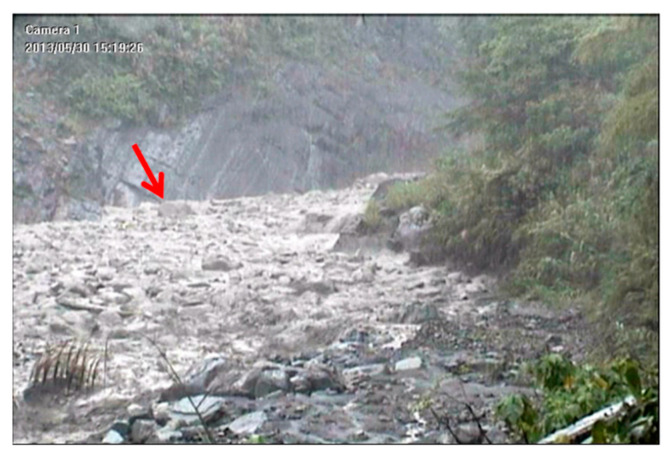
Image taken by the CCD camera deployed at Aiyuzi Stream at 15:19:26 on 30 May 2013. The stone indicated by the red arrow has an estimated diameter of 0.5 m.

**Figure 18 sensors-22-09310-f018:**
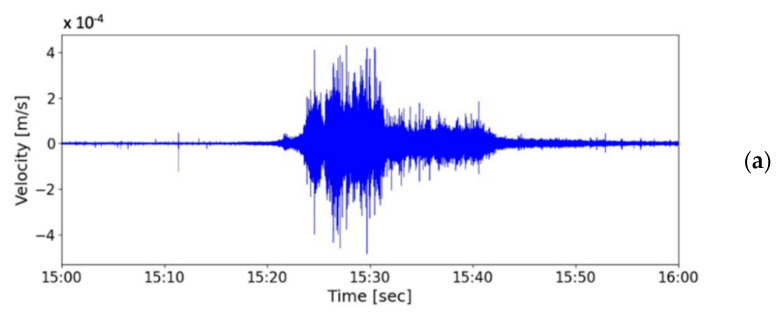
(**a**) Time-series data of ground vibration in the z-axis measured by the upstream seismometer at Aiyuzi Stream from 15:00 to 16:00 (LT) on 30 May 2013. (**b**) Spectrogram of these signals. Colors on the PSD indicate vibration strength.

**Figure 19 sensors-22-09310-f019:**
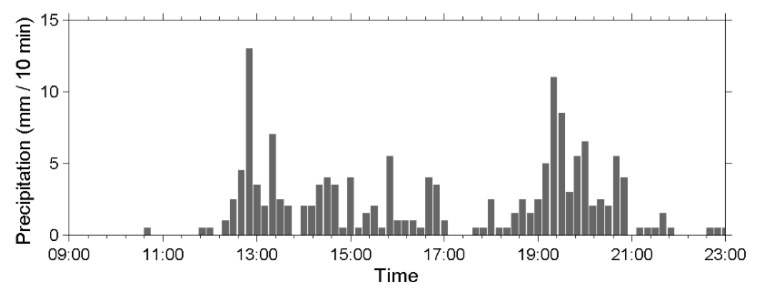
Precipitation chart measured in 10 min intervals measured by the rain gauge at Shen-Mu Station on 20 May 2014.

**Figure 20 sensors-22-09310-f020:**
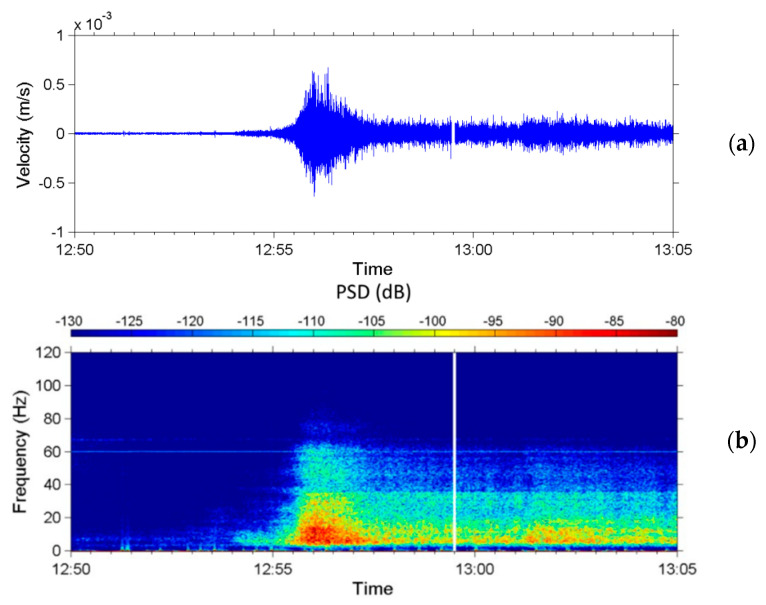
(**a**) Time-series data of ground vibrations in the z-axis measured by the upstream seismometer at Aiyuzi Stream from 12:50 to 13:05 (LT) on 20 May 2014. (**b**) Spectrogram of these signals.

**Figure 21 sensors-22-09310-f021:**
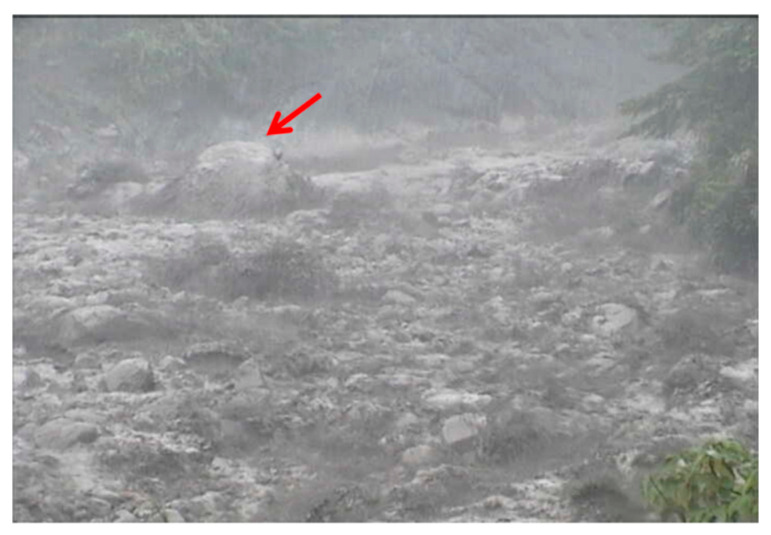
Image taken by the CCD camera deployed at Aiyuzi Stream at 12:54:19 on 20 May 2014. The stone indicated by the red arrow has an estimated diameter of 2.0 m.

**Figure 22 sensors-22-09310-f022:**
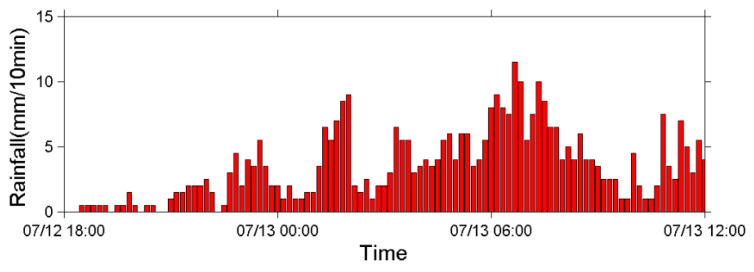
Precipitation chart measured at 10 min intervals by the rain gauge at the Shen-Mu Station during Typhoon Soulik.

**Figure 23 sensors-22-09310-f023:**
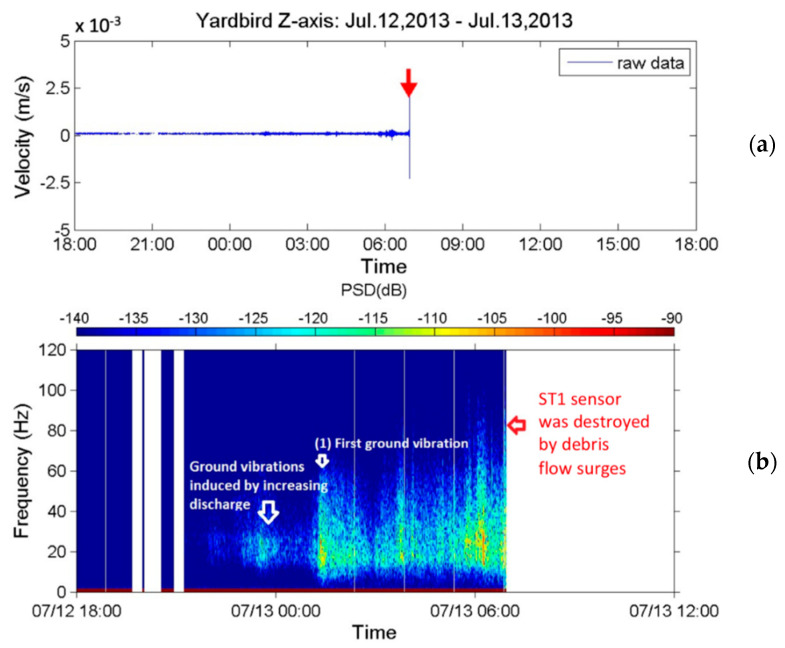
Seismic signals in the z-axis measured by the upstream seismometer at Aiyuzi Stream. (**a**) Time-series data from 18:00 on 12 July to approximately 07:00 on 13 July and (**b**) corresponding spectrogram. Detailed spectrograms from (**c**) 01:00 to 02:00 and (**d**) 06:51 to 06:58 on 13 July. Colors on the PSD indicate vibration strength. The red arrow in subfigure (**a**) marks the time when the upstream seismometer was destroyed.

**Figure 24 sensors-22-09310-f024:**
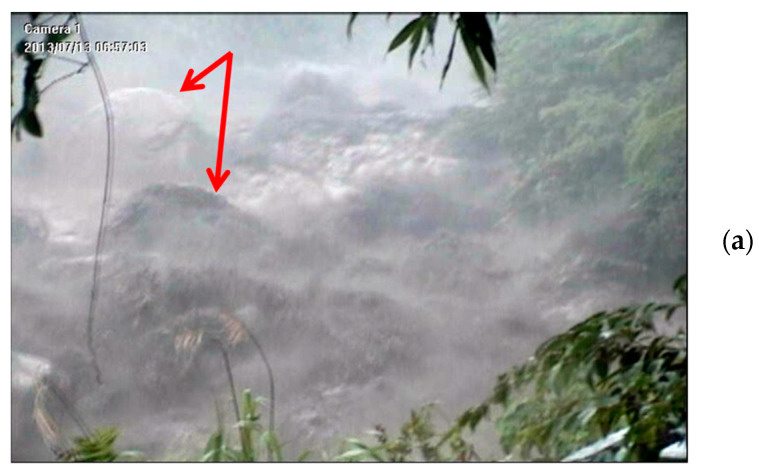
Images taken by the CCD camera deployed at the Aiyuzi Stream on 13 July 2013 (**a**) at 06:57:03, when the debris flow surge was approaching the upstream seismometer and (**b**) at 06:57:28 when the debris flow surge passed the CCD camera. The stones indicated by the red arrows were estimated to have a diameter of 3–5 m.

**Figure 25 sensors-22-09310-f025:**
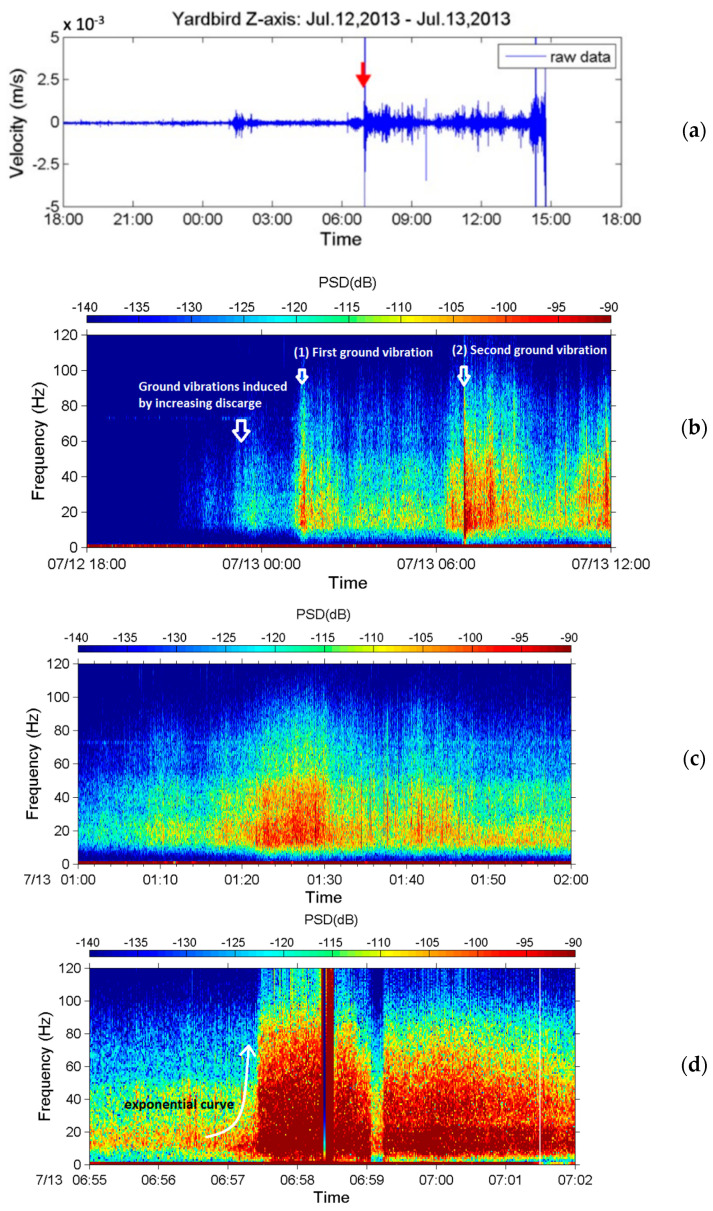
Seismic signals in the z-axis measured by the downstream seismometer at Aiyuzi Stream; (**a**) time-series data from 18:00 on 12 July to approximately 15:00 on 13 July. Spectrograms of the seismic signals from (**b**) 18:00 on 12 July to 12:00 on 13 July, (**c**) 01:00 to 02:00 on 13 July, and (**d**) 06:51 to 06:58 on 13 July. The colors on the PSD indicate vibration strength. The red arrow in subfigure (**a**) marks the time when the second debris flow surge occurred at 06:57:17.

**Figure 26 sensors-22-09310-f026:**
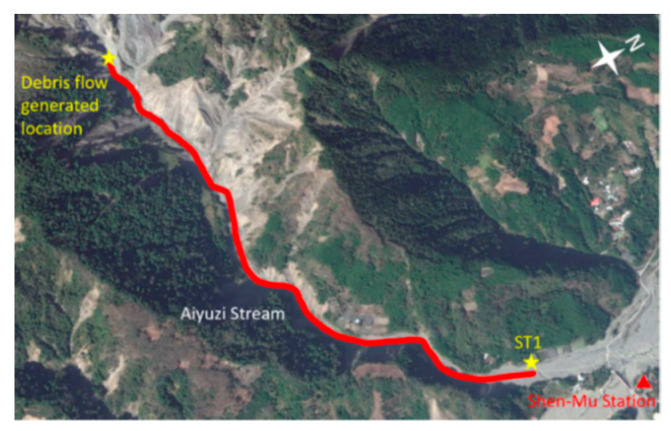
Approximate location at which the debris flows might have been generated (from Google Earth).

**Table 1 sensors-22-09310-t001:** Characteristics of Yardbird seismometer and various seismic measurement devices for comparison.

	Broadband Seismometer (STS-2)	Geophone (GS-20DX)	Raspberry Shake 3D	LE-3Dlite	Yardbird Seismometer
Working frequency range (Hz)	0.0083–100	8–250	0.5–40	1–100	0.3–120
Sensitivity (V/m/s)	1500	27.6	75	800	150
Weight (kg)	13	0.08	0.6	1.9	1.9

**Table 2 sensors-22-09310-t002:** Correlation between observed rock size and the lowest observed frequency.

Event	Date	Debris Flow Scale	Rock Size in the Surge Front (m)	Lowest Frequency * (Hz)
1	30 May 2013	small	0.5	11
2	13 July 2013	large	3–5	2
3	20 May 2014	medium	2.0	5

* Lowest frequency at the time when the rock images were taken.

**Table 3 sensors-22-09310-t003:** Main frequency ranges of seismic signals produced by various geophysical events.

Events	Specifications	Frequency Range (Hz)	References
Earthquakes	Local (~50 km) Regional (~230 km) Teleseism (~7000 km)	1–50 1–12 0.1–1	Suriñach et al. [42]
Rockfalls	masses 76–472 kg masses 466–13,581 kg	5–120	Hibert et al. [38] Saló et al. [39]
Volcanic eruptions	Popocatépetl Volcano	0.1–10	Arámbula-Mendoza [50] Matoza et al. [51]
Debris flows	Large boulders in the surge front	2–120	Present Study Lai et al. [18], Schimmel et al. [14], Marchetti et al. [19], Belli et al. [20]
Landslides	Laguna Beach 2005 Shiaolin landslide 2009	1–10 0.5–1.5	Suriñach et al. [42] Feng [52]
Snow avalanches	Small scale Large scale	2–45 0.1–90	Suriñach et al. [42]
River flows		10–30, 8–70 2–80	Present Study Burtin et al. [47,48]

## Data Availability

Not applicable.

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
