# Peer review of "Low-Frequency Ground Vibrations Generated by Debris Flows Detected by a Lab-Fabricated Seismometer"

_sensors, 2022, doi:10.3390/s22239310_

Round 1
Reviewer 1 Report
For the lab-fabricated seismometer was able to detect one debris flow approximately 3 min and 40 s before it arrived, the work is of some significance to debris flow hazards early warning.
1. Line 45. “;” -> “,” .
2. Lines 640-644. For “Many geophysical events produce seismic waves with frequencies ranging from 0.3 to 120 Hz, which lie within the working frequencies of the Yardbird seismometer. These events include earthquakes, volcanic eruptions, rockfalls, snow avalanches, debris flows, pyroclastic flows, and landslides, and could be identified using detected signals based on their characteristic frequency ranges.” How can you distinguish different geophysical events when their frequency ranges are overlapped ? For example, how can you separate debris flows from river flows when their frequencies are all 50Hz according to Table 3?
3. References are too old. Please add 5-10 references published in the latest 3 years.
Author Response
Response to Reviewer 1’s comments:
We appreciate the interest that the reviewers have taken in our manuscript and the constructive suggestions they have provided. Our revisions reflect all the reviewers’ suggestions and comments. Detailed responses are provided below.
- Line 45. “;” -> “,”.
Reply:
“;” in Line 45 has been changed to “,”.
- Lines 640-644. For “Many geophysical events produce seismic waves with frequencies ranging from 0.3 to 120 Hz, which lie within the working frequencies of the Yardbird seismometer. These events include earthquakes, volcanic eruptions, rockfalls, snow avalanches, debris flows, pyroclastic flows, and landslides, and could be identified using detected signals based on their characteristic frequency ranges.” How can you distinguish different geophysical events when their frequency ranges are overlapped? For example, how can you separate debris flows from river flows when their frequencies are all 50Hz according to Table 3?
Reply:
Distinguishing geological events with overlapping seismic frequency ranges is beyond the scope of this study. However, Table 3 indicates that the seismic signals induced by river flows typically have a smaller frequency range of 8–70 Hz, whereas those of debris flows have a wider range of 2–120 Hz. Moreover, events can be distinguished by variables other than frequency range, such as signal duration or PSD strength. We added a new reference (Cook and Dietze [52]) that discusses the characteristics of seismic signals produced by various geophysical events, such as rockfalls, river flows, floods, landslides, and earthquakes.
- References are too old. Please add 5-10 references published in the latest 3 years.
Reply:
We have added five new references published in 2021–2022 to the revised manuscript with accompanying text as follows.
Tiwari et al. [29] discussed the seismic characteristics of a debris flow caused by a rock-ice avalanche occurring in the Himalayan glaciers. (Lines 69-70)
Cook et al. [30] reported that the detection and early warning of catastrophic flow events can be achieved using a regional seismic monitoring system. (Lines 70-72)
Recently, Cook and Dietze [52] provided a detailed discussion of the characteristics of seismic signals produced by geomorphic processes, such as rockfalls, river flows, floods, landslides, and earthquakes. (Lines 657-659)
Additionally, Schimmel et al. [23] and De Haas et al. [24] have been listed as references [23-24] and mentioned in Lines 56-57.
New References:
[23] Schimmel, A.; Coviello, V.; Comiti, F. Debris flow velocity and volume estimations based on seismic data. Nat. Hazards Earth Syst. Sci. 2022, 22, 1955-1968, doi.org/10.5194/nhess-22-1955-2022.
[24] De Haas, T.; Åberg, A.S.; Walter, F.; Zhang, Z. Deciphering seismic and normal-force fluctuation signatures of debris flows: An experimental assessment of effects of flow composition and dynamics. Earth Sur. Process. Landforms 2021, 46, 2195-2210.
[29] Tiwari, A.; Sain, K.; Kumar, A.; Tiwari, J.; Paul, A.; Kumar, N.; Haldar, C.; Kumar, S.; Pandey, C.P. Potential seismic precursors and surficial dynamics of a deadly Himalayan disaster: an early warning approach. Sci. Rep. 2022, 12, 3733, doi.org/10.1038/s41598-022-07491-y.
[30] Cook, K. L.; Rekapalli, R.; Dietze, M.; Pilz, M.; Cesca, S.; Purnachandra Rao, N.; Srinagesh, D.; Paul, H.; Metz, M.; Mandal, P.; Suresh, G.; Cotton, F.; Tiwari, V.M.; Hovius, N. Detection and potential early warning of catastrophic flow events with regional seismic networks. Sciences 2021, 374, 87-92.
[52] Cook, K.L.; Dietze, M. Seismic Advances in Process Geomorphology. Annu. Rev. Earth Planet. Sci. 2022, 50, 183-204.

Reviewer 2 Report
This manuscript introduced a lab-fabricated ocean bottom seismometer and deployed terrestrially to detect low-frequency ground vibrations produced by debris flows. However, there are still some questions and suggestions for this study.
1. At the second last paragraph of introduction section, the authors should elaborate a clear purpose and scope of your work. What is the difference between the previous works and yours? What is the significance of this work?
2. Please improve the quality of some figures. Particularly, Figures 6a, 7a and 8a. It is difficult to identify the lines in b/w in print.
3. In the Conclusion section, it is recommended to address the more contributions of this work.
4. In many places, the text is unclear and the English language used is not at high quality. Please make them more concise.
Author Response
Response to Reviewer 2’s comments
We appreciate the interest that the reviewers have taken in our manuscript and the constructive suggestions they have provided. Our revisions reflect all the reviewers’ suggestions and comments. Detailed responses are provided below.
- At the second last paragraph of introduction section, the authors should elaborate a clear purpose and scope of your work. What is the difference between the previous works and yours? What is the significance of this work?
Reply:
The following text has been added to the end of the second last paragraph of the introduction section to clarify the difference between the present work and previous works as well as its significance.
“In previous studies, commercially available seismometers have been employed to detect low-frequency ground vibrations [18-20]. In contrast to commercially available seismometers, lab-fabricated seismometers can be customized to measure various frequency ranges depending on the needs of the user. The results of this study could provide useful information and techniques for developing remote-sensing and early-warning devices for various geophysical events.” (Lines 132-137)
- Please improve the quality of some figures. Particularly, Figures 6a, 7a and 8a. It is difficult to identify the lines in b/w in print.
Reply:
Figures 6a, 7a, and 8a present the seismic signals obtained from the geophone at various measuring distances. Because the sensitivity of geophone is lower than that of the Yardbird seismometer (corresponding results obtained from the Yardbird seismometer are presented in Figures 6b, 7b, and 8b), the geophone’s signal-to-noise ratio is also lower. Therefore, as the source–sensor distance increased, the signals detected by the geophone are increasingly less clear than those detected by the Yardbird seismometer. The signals in Figures 6a, 7a, and 8a are primarily measured ambient noise. Because the sampling rate was high (240 Hz), the measured ambient noises have a zigzag appearance, as shown in these figures. Unfortunately, the resolution cannot be improved.
- In the Conclusion section, it is recommended to address the more contributions of this work.
Reply:
We have added the following text to the end of the Conclusion section to further describe the contributions of this work.
“Overall, the results of this study reveal that the Yardbird seismometer is a cost-effective sensor for both debris flow research and implementing early warning systems for geophysical events. Future studies could further examine its performance for detecting other events, such as rockfalls, landslides, or earthquakes.” (Lines 701-704)
- In many places, the text is unclear and the English language used is not at high quality. Please make them more concise.
Reply:
The English writing in some parts of the old manuscript have been improved by an English editing expert. For example, the following texts have been revised.
- Old version:
“The two wire sensors were deployed at the same location with the lower one is approximately 50 cm above the river bed and the upper one is 100 cm higher than the lower one. Distance between wire sensor and CCD camera was 100 m and that between the CCD camera and the upstream seismometer (ST1) was approximately 30 m.”
New Version:
“The two wire sensors were deployed at the same location at approximately 50 and 150 cm above the river bed. The distance between the wire sensors and the CCD camera was 100 m, and that between the CCD camera and the upstream seismometer (ST1) was approximately 30 m.” (Lines 279-282)
- Old version:
“The continuous signal at 60 Hz is attributable to the nearby poorly-isolated cable of alternating current.”
New Version:
“The continuous signal at 60 Hz is attributable to a nearby poorly-isolated cable carrying alternating current.” (Lines 378-380)

Reviewer 3 Report
L.035-036
"debris flow" is dublicated.
L.451
urge -> surge?
L.611-612
The velocity of 11.8 m/s is an instant value and there is no evidence suggesting this velocity corresponds to the averave velocity. Therefore the estimated distance from the sensor to the source is highly uncertain. The author should comment it.
Author Response
Response to Reviewer 3’s comments
We appreciate the interest that the reviewers have taken in our manuscript and the constructive suggestions they have provided. Our revisions reflect all the reviewers’ suggestions and comments. Detailed responses are provided below.
- 035-036, "debris flow" is duplicated.
Reply:
“debris flows” in Line 36 has been deleted.
- 451, urge -> surge?
Reply:
“After the debris flows urge…” in Line 450 has been changed to “After the debris flow surge…”. (Line 457 in the new manuscript)
- 611-612, The velocity of 11.8 m/s is an instant value and there is no evidence suggesting this velocity corresponds to the average velocity. Therefore, the estimated distance from the sensor to the source is highly uncertain. The author should comment it.
Reply:
As mentioned in the text (Lines 602-604), a velocity of 11.8 m/s was obtained by dividing the distance between the two sensors (200 m) by the time required for the debris flow to cross this distance (17 s). This velocity is not instantaneous velocity; but mean velocity. However, we do agree with the reviewer’s comment that the estimated distance from the upward sensor to the debris flow source is uncertain because it was estimated based on the debris flow velocity between the two sensors, which were deployed far from the flow source location. The following text has been added to address the reviewer’s concern.
“Because the flow source was estimated on the basis of the averaged debris flow velocity between the two seismometers, which were both deployed far from the flow source, this location is only a rough estimate.” (Lines 621-623)

Round 2
Reviewer 2 Report
The authors have satisfactorily addressed the concerns of reviewer.